# Inverse Design of Microstructures via Generative Networks for Organic Solar Cells

**Xian Yeow Lee**    **Aditya Balu**    **Balaji Sesha Sarath Pokuri**
**Adarsh Krishnamurthy**    **Soumik Sarkar**    **Baskar Ganapathysubramanian†**

Iowa State University
†baskarg@iastate.edu

## Abstract

We consider the inverse problem of efficiently designing material microstructures that exhibit desired electrical properties in an organic solar cell design. We leverage data-driven generative models to learn the underlying data distribution and generate novel microstructures during test time. We focus on a recent framework, specifically generative invariance networks (InvNets), which simultaneously learns from a dataset of microstructures while constraining the output of the generative model to conform to constraints such as generating microstructure designs with a targeted short circuit current density, $J$ values. While previous works in this area have focused on the model training and data efficiency aspects, the applicability and success of Generative Invariance Networks to different material systems (i.e., the donor material and the acceptor material chemistry) and device thickness remain unexplored. In this paper, we demonstrate that we can successfully adapt the same InvNet framework to different material systems and device thicknesses with minimal computational effort.

## Introduction

Our overarching objective is to generate two-phase microstructures that exhibit a specific (large) value of short circuit current density, $J$, as shown in Fig. 1. Existing works have applied variants of the generative invariance network (InvNet) framework to design (or rather 'reconstruct') microstructures exhibiting desired properties like volume fraction and 2-point correlations (Joshi et al. 2020). However, the type of invariances considered was limited to continuous and differentiable functions to obtain useful gradient information to optimize the generative model's parameters. To extend the framework to generate microstructures that exhibit desired photovoltaic properties such as $J$, which is both computationally slow to calculate and non-differentiable, Lee et al. (2021) proposed to train a neural-network-based surrogate model to map the generated microstructures to $J$ to compute the loss and obtain surrogate gradient directions to update the generative model. An expected bottleneck of training a surrogate model on high-fidelity full physics simulations labels is the challenge of data generation, which may be computationally expensive. Various works in literature have previously explored the idea of leveraging both high- and low-fidelity

data to accelerate computational models (Sarkar et al. 2019; Costabal et al. 2019; Babaee et al. 2020; Liu and Wang 2019; Tao and Sun 2019; Zhou et al. 2021; Guo et al. 2018).

Similarly, Lee et al. (2021) exploit a similar idea by proposing and incorporating a multi-fidelity surrogate model into the InvNet framework to circumvent the challenge of generating computationally expensive high-fidelity labels. This approach has produced promising results and exhibited the ability to generate microstructures on demand, satisfying a user-defined value of $J$. Nevertheless, while successful, the applicability of the proposed InvNet framework with a surrogate model to generate microstructures from different material systems and device thickness remains unexplored. In other words, we seek to validate if the generative InvNet framework is generalizable to more than one material system and OPV device thickness. We highlight that the notion of generalizability we explore is in terms of architecture generalizability, rather than generalizability of the trained model. As such, our work contributes to the following: We demonstrate that InvNet's architecture **1)** is generalizable for different material systems, **2)** is generalizable for different device thickness, **3)** works for generating microstructures with targeted $J$ even with only a 1000 samples to train the multi-fidelity surrogate model ($\approx$ 7X lesser than the original amount of data used to train the multi-fidelity surrogate model).

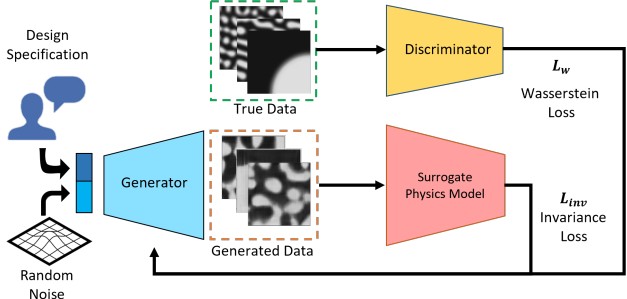

Figure 1: Overview of the Generative Invariance Network framework with a multi-fidelity surrogate physics model as an invariance checker. The surrogate model is parameterized by a neural network that predicts the $J$ value of a microstructure. Figure adapted from Lee et al. (2021).

Table 1: Parameters used for the simulation of P3HT-PCBM and PBDB-T-ITIC material system

| Name | P3HT-PCBM | PBDB-T-ITIC |
|---|---|---|
| Electron zero-field mobility $(m^2V^{-1}s^{-1})$ | $6 \times 10^{-6}$ | $1 \times 10^{-6}$ |
| Hole zero-field mobility $(m^2V^{-1}s^{-1})$ | $1.4 \times 10^{-6}$ | $2 \times 10^{-6}$ |
| Exciton mobility $(m^2V^{-1}s^{-1})$ | $2 \times 10^{-7}$ | $2 \times 10^{-7}$ |
| HOMO energy of P3HT $(eV)$ | $-4.65$ | $-5.3$ |
| LUMO energy of PCBM $(eV)$ | $-3.8$ | $-3.8$ |
| Relative dielectric constant of donor | $3.0$ | $2.8$ |
| Relative dielectric constant of acceptor | $3.9$ | $4.5$ |
| Average exciton lifetime $(s)$ | $0.4 \times 10^{-9}$ | $1.38 \times 10^{-9}$ |
| Electron-hole separation distances $(m)$ | $1.8 \times 10^{-9}$ | $1.8 \times 10^{-9}$ |

## Methods

### Data generation

To investigate InvNet's capability to generalize towards a different material system and device thickness, we first explore two different material systems, specifically P3HT-PCBM and PBDB-T-ITIC. For our experiments, we sampled 1000 samples from the microstructure dataset that was developed by solving the Cahn-Hillard equations (Cahn and Hilliard 1958) to generate binary microstructures with varying morphologies. Using these microstructure samples, we performed a full-physics simulation to compute the high-fidelity $J$ values for both material systems. These full-physics simulations were conducted using an in-house solver and involved solving the Excitonic-Drift-Diffusion equations to obtain the $J$ values based on the morphology of the microstructure and constants values that are dependent on the material system. The material-specific parameters that are required to perform the full physics simulations for both P3HT-PCBM and PBDB-T-ITIC are given in Table 1. In addition to these parameters, we set the device thickness to be *100nm*. While each morphology is provided as a $101 \times 101$ pixel image, the full physics simulation required discretizing each morphology into a mesh of size $\sim 512 \times 512$ to capture the steep gradients in the field solutions. We ran each simulation on a high-performance computing cluster, with the generation of each sample taking about 2-5 minutes for performing the simulation using 32 CPU cores. Additionally, we also generated a third set of $J$ values by setting the device thickness values to *200nm* for the P3HT-PCBM material system. This effectively generates a different set of $J$ values for the same morphologies but represent different device thicknesses.

### Training the multi-fidelity surrogate model

Following the training procedures described in Lee et al. (2021), we trained the multi-fidelity surrogate model that estimates $J$ using a limited amount of high-fidelity labels, specifically only 1000 labels and additional low-fidelity labels. The multi-fidelity surrogate model, illustrated in Fig. 2, consists of two parallel neural networks. The first network (shared embedding network) is made up of convolutional layers that map a microstructure to an embedding vector, $h \in \mathbb{R}^{16}$. The second network (low-fidelity network) is a pre-trained convolutional network that maps the microstructure to a set of low-fidelity labels, $g \in \mathbb{R}^3$. Here, the low-fidelity

network was pre-trained a dataset of 38K morphologies and corresponding 38k labels of $g$. These low-fidelity labels were computed using a cheap, graph-based approach that maps an input morphology to a performance metric that was shown to be weakly correlated with $J$ (Wodo et al. 2012). Additionally, we highlight that the 38k morphologies and their corresponding $g$ labels do not change with microstructure or the device thickness. However, the high-fidelity labels $J$ do need to be evaluated for the change in the material system and device thickness. The outputs of the shared embedding network, $h$ and low-fidelity network, $g$, are then concatenated and sent through a third neural network consisting of two fully-connected layers to predict the $J$ value of a morphology. We train the surrogate models from scratch with the three sets of simulated ground truth $J$ values using the mean square error loss and optimized via conventional optimizers.

### Training InvNet

To train InvNet, we retain the same model architecture and training procedure as described in Lee et al. (2021) for a fair comparison. Specifically, the InvNet consists of a Wasserstein-GAN (Gulrajani et al. 2017), which is composed of a generator and a discriminator, both parameterized by convolutional neural networks. During training, the generator takes as input a batch of random vector $z$ and a batch of random design specification $j \in \mathbb{J}$, where $\mathbb{J}$ is the set of all valid $J$ values and outputs a $128 \times 128$ image of a microstructure. The discriminator compares the generated images with a batch of true images sampled from the Cahn-Hillard dataset and outputs the Wasserstein distance signaling if the distribution of images generated matches the true data distribution. In addition to that, the trained multi-fidelity model is also used as an invariance model that checks if the generated microstructures properties match the design specification $j$. Both the generator and discriminator are optimized sequentially until the model converges. Note that the multi-fidelity surrogate model's weights are not updated during training and are purely used as a function evaluator to estimate the $J$ value of any generated microstructures.

## Results and Discussion

### P3HT - PCBM material system for *100nm* device

Using the full-physics labels generated, we trained the multi-fidelity network described in the previous section to predict the $J$ values on a held-out test set of microstructures, which resulted in a $R^2$ of 0.974 for $J$. Fig. 3 shows the scatter plot of the property estimated by the multi-fidelity model against the ground truth values. As observed, the multi-fidelity model was capable of predicting the $J$ accurately. We stress that here we train the multi-fidelity model with only 1000 high-fidelity labels as opposed to 20% (7.6k labels) of the entire dataset (38k labels), which was the number of labels originally used to train the multi-fidelity model in Lee et al. (2021).

Next, we present the results of training *InvNet* with the multi-fidelity surrogate model framework. In Fig. 4(a), we show samples of microstructures generated with InvNet from low $J$ values (top row) to high $J$ values (bottom row). We observe that the InvNet framework can generate structurally

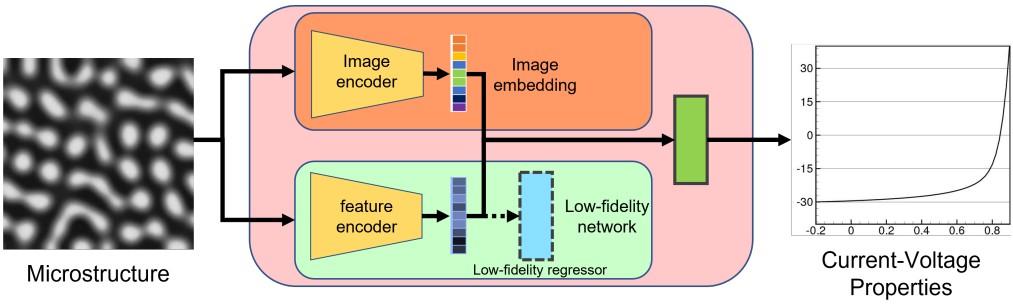

Figure 2: Architecture of multi-fidelity surrogate model used to predict $J$ given a morphology using 1000 high-fidelity labels and a low-fidelity network that was pre-trained on 38k low-fidelity labels.

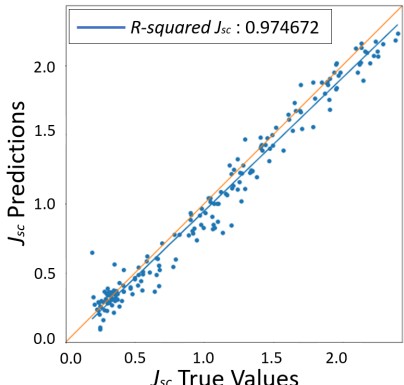

Figure 3: Correlation plot of the estimated $J$ for P3HT-PCBM material system using the multi-fidelity surrogate physics model with respect to the ground truth values.

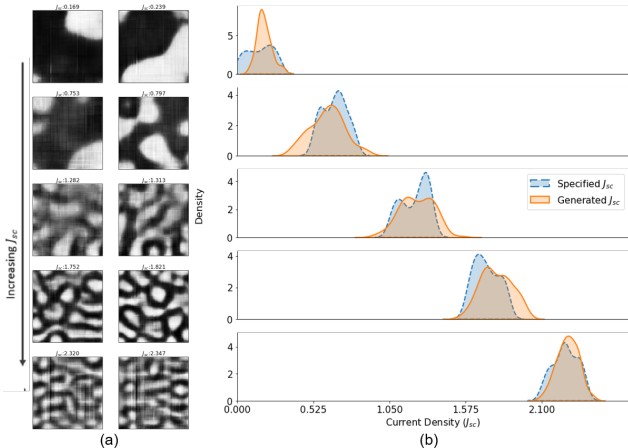

Figure 4: **(a)** Examples of microstructures generated by InvNet for various $J$ values for P3HT-PCBM material system. **(b)** Densities of predicted $J$ from generated morphologies compared with a range of specified $J$ for 1000 samples.

different microstructures as we change the design specifications. Additionally, we also observe no evidence of mode collapse because the microstructures in the same row (similar $J$ value) have significantly different structures. To further analyze the design specification satisfaction of the generated microstructures beyond just two anecdotal examples, we used InvNet to generate another 1000 microstructures for different ranges of $J$ and used the multi-fidelity surrogate model to predict the $J$ values of these generated microstructures. The distribution of the design specifications and predicted $J$ values for the generated microstructures are shown in Fig. 4(b). Based on the highly overlapping distributions of input specification and output property, we confirm that InvNet can generate unique microstructures that conform to different values of specified $J$. These observations also aligned with the findings by Lee et al. (2021).

## Performance of InvNet on different a material system and device thickness

Next, we present the results for training the multi-fidelity surrogate model and InvNet for PBDB-T-ITIC material system and P3HT-PCBM with device thickness of *200nm*. The correlation plots for training the multi-fidelity surrogate for PBDB-T-ITIC and for P3HT-PCBM with device thickness

of *200nm* are shown Fig. 5 and Fig. 6 respectively. We observe that the performance of the regressor for PBDB-T-ITIC material system has a good correlation ($R^2 = 0.96$) between the predicted $J$ values and the actual simulated $J$ values. For the device thickness of *200 nm*, we obtain a marginally lower correlation ($R^2 = 0.89$). This may be attributed to the fact that the distribution exhibited by the $J$ (the limited high-fidelity labels) for *200 nm* thickness devices used in training the model is shifted from the actual distribution exhibited by the full data.

Using these two newly trained surrogate models mentioned above as the invariant module, we trained two more InvNet to generate new microstructures for the different material systems and device thickness. Fig. 7 and Fig. 8 show the morphologies generated for PBDB-T-ITIC material system and the device thickness of *200 nm* for P3HT-PCBM material system. In both scenarios, we observe InvNet can generate diverse microstructures that exhibit different structures as we vary the $J$ specifications. Furthermore, we observe that the distribution of $J$ values of the generated microstructures also matches the distribution of specified $J$ values, thus demonstrating that the InvNet framework is generalizable to

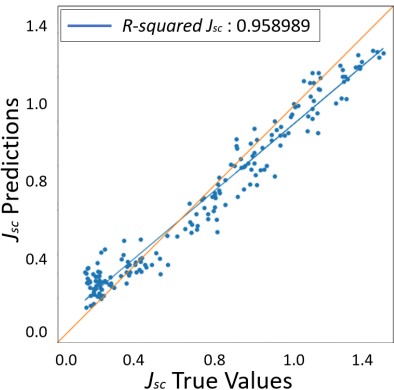

Figure 5: Surrogate physics model correlation plot for PBDB-T ITIC material system with same device thickness of 100nm.

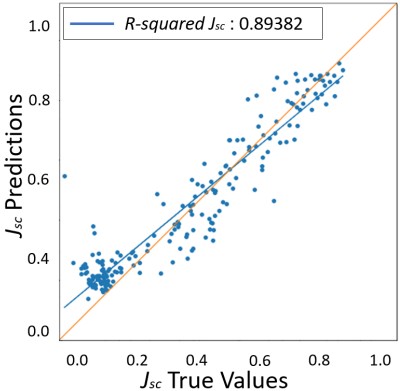

Figure 6: Surrogate physics model correlation plot for P3HT-PCBM material system with a device thickness of 200nm.

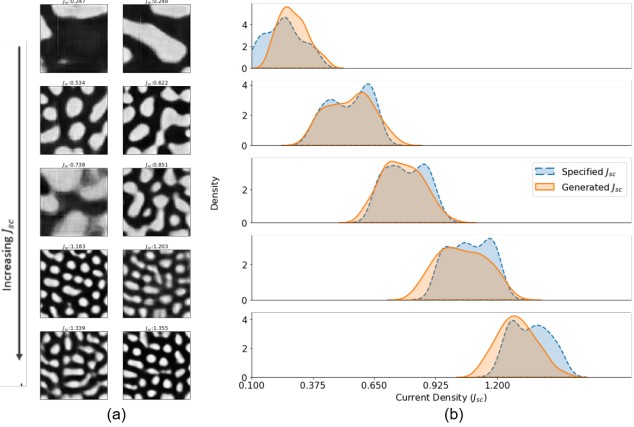

Figure 7: $J$ values for PBDB-T-ITIC material system. (b) Densities of predicted $J$ from generated morphologies compared with a range of specified $J$ for 1000 samples.

different material systems and device thickness.

Lastly, in Fig. 9, we contrast various morphology designs

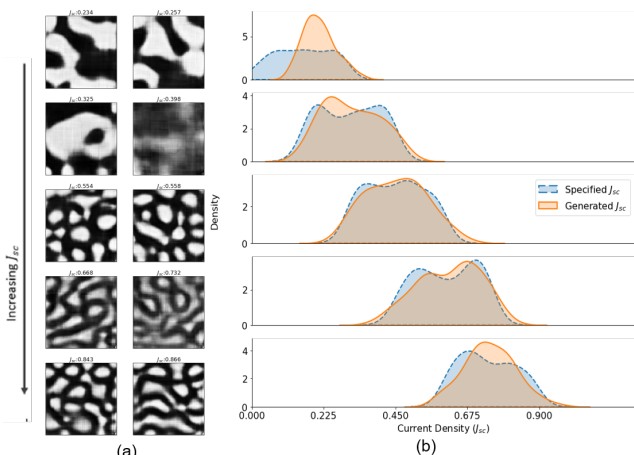

(a)                    (b)

Figure 8: (a) Examples of morphologies generated by InvNet for the specified $J$ values for P3HT-PCBM material system with a device thickness of *200 nm*. (b) Densities of predicted $J$ from generated morphologies compared with a range of specified $J$ for 1000 samples.

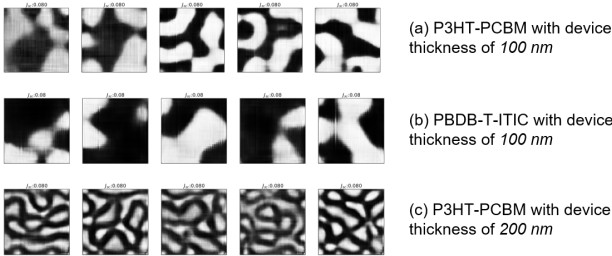

(a) P3HT-PCBM with device thickness of *100 nm*

(b) PBDB-T-ITIC with device thickness of *100 nm*

(c) P3HT-PCBM with device thickness of *200 nm*

Figure 9: Different morphologies generated for a fixed $J_{sc}$ value, but different random seeds of the generator and different material configurations.

generated by InvNets for different material systems given a fixed design specification of $J$. Each row represents the sample generated for a specific material system, and each column represents an instance of the sample generated using a fixed random vector $z$. As observed, the unique microstructures obtained for each material system is very diverse, and multiple possible designs do exist for a given $J$ value. This is beneficial as it provides OPV manufacturers with multiple design options to satisfy manufacturing constraints rather than optimizing for a single morphology design.

## Conclusions

We explore the generalizability of Generative Invariance Networks (InvNet), a data-driven constrained generative model, to generate novel microstructure designs for different material systems and designs. Our results validate that the InvNet framework and architecture can generalize to data derived from different material systems and device thicknesses. Future work will explore methods to improve the models' performance as well as extendability of this framework to three-dimensional designs of microstructures and an framework that also incorporates material chemistry as an input.

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
