# OpenReview forum: "Inverse Design of Microstructures via Generative Networks for Organic Solar Cells"
_AAAI.org/2022/Workshop/ADAM — AAAI 2022 Workshop ADAM_

### Official Review · Reviewer_niei · 2021-11-24
**Well-written paper. Interesting inverse design model with sound preliminary results of 2D microstructure designs. Recommend it to be accepted.**

**Rating:** 9
**Confidence:** 5

**Review:**

Inverse design is an interesting topic. It short-circuits the costly iteration of gradient-based design optimization by directly mapping performance metrics to design variables. This paper maps the target short circuit current density J to the 2D microstructure designs represented as images. A generative model, InvNets, is used to learn the mapping by constraining J as the invariance of the generative model. The paper is well written with sound and promising preliminary results.

Pros: Well-written paper. Clear framework and methodology. Promising results showing the impact to the domain of organic solar cell design.

Cons: Not clear how the surrogate simulator is used in the InvNets training given the surrogate model's weights are not updated during the training. It looks like the output of the surrogate model is again fed into the generator.

---

### Official Review · Reviewer_EaMB · 2021-11-29
**Generalizability of InvNet across material system and device thickness**

**Rating:** 6
**Confidence:** 4

**Review:**

This paper applies the InvNet framework developed by some of the authors in a previous paper to another material system and a different device thickness, showing the framework generalizes. Since the model incorporates a physics based surrogate, I expected the authors would investigate generalizability without retraining the InvNet and surrogate model on different material and/or thickness, however the description seems to suggest both are retrained - if this is not the case, please highlight that. If retraining was indeed done, then is the paper claiming generalizability of the architecture only? If so, this should be clarified. Also, was the retraining done in a 'fine-tuning' manner or starting from scratch? The obtained density profiles are good matches to the target - could this match be improved further with additional iterations or was a local minimum obtained - please comment. It is interesting that no mode collapse was observed at least for some of the target densities. It would be interesting to see the diversity of J profiles generated by different initializations in Fig 9.